# Using Zebrafish as a Disease Model to Study Fibrotic Disease

**DOI:** 10.3390/ijms22126404

**Published:** 2021-06-15

**Authors:** Xixin Wang, Daniëlle Copmans, Peter A. M. de Witte

**Affiliations:** 1Laboratory for Molecular Biodiscovery, Department of Pharmaceutical and Pharmacological Sciences, KULeuven-University of Leuven, O&N II Herestraat 49-Box 824, 3000 Leuven, Belgium; xixin.wang2020@outlook.com (X.W.); danielle.copmans@kuleuven.be (D.C.); 2Biology Institute, Qilu University of Technology (Shandong Academy of Sciences), 28789 East Jingshi Road, Jinan 250103, China

**Keywords:** zebrafish, animal models, fibrotic disease, chemical induction, genetic manipulation, ECM accumulation

## Abstract

In drug discovery, often animal models are used that mimic human diseases as closely as possible. These animal models can be used to address various scientific questions, such as testing and evaluation of new drugs, as well as understanding the pathogenesis of diseases. Currently, the most commonly used animal models in the field of fibrosis are rodents. Unfortunately, rodent models of fibrotic disease are costly and time-consuming to generate. In addition, present models are not very suitable for screening large compounds libraries. To overcome these limitations, there is a need for new in vivo models. Zebrafish has become an attractive animal model for preclinical studies. An expanding number of zebrafish models of human disease have been documented, for both acute and chronic diseases. A deeper understanding of the occurrence of fibrosis in zebrafish will contribute to the development of new and potentially improved animal models for drug discovery. These zebrafish models of fibrotic disease include, among others, cardiovascular disease models, liver disease models (categorized into Alcoholic Liver Diseases (ALD) and Non-Alcoholic Liver Disease (NALD)), and chronic pancreatitis models. In this review, we give a comprehensive overview of the usage of zebrafish models in fibrotic disease studies, highlighting their potential for high-throughput drug discovery and current technical challenges.

## 1. Introduction

Fibrotic disorders, including various cardiovascular diseases (CVD), liver cirrhosis and chronic kidney disease (CKD), are characterized by a progressive and irreversible accumulation of extracellular matrix in the organs concerned. Eventually, this excess of fibrotic tissue impairs the function of the organ, potentially resulting in a life-threatening situation [1,2,3]. Fibrotic diseases affect numerous people, in steadily increasing numbers. For instance, according to WHO global health estimates, around 31% of all deaths worldwide are attributed to CVD each year, and cardiac fibrosis (CF) is implicated in almost all forms of CVD [4]. Furthermore, 2.1% of all deaths were caused by liver cirrhosis in 2015 [3], and 1.5% of deaths were attributable to CKD in 2012 [1]. In fact, it is difficult to accurately estimate the incidence of each fibrotic disease, because most of these disorders are concealed at early stages and manifest themselves across multiple organ systems. One documented estimation is that chronic fibrotic diseases contribute to as much as 45% of all-cause mortality collectively, in developed countries [5,6].

In the past decades, many in vivo rodent models of fibrotic diseases have been generated, and consequently used to study their mechanistic underpinnings. These models have also significantly contributed to the discovery of antifibrotics. Some compounds have been translated from bench to bedside for hepatic fibrosis (e.g., GR-MD-02, GM-CT-01) [7], and others have even obtained market authorization for idiopathic pulmonary fibrosis treatment (e.g., nintedanib, pirfenidone) [8].

However, although these rodent models often possess a high construct validity, closely mirroring the human disease, their generation is time-consuming. In addition, present models are not very suitable for screening of large compounds of libraries owing to their high costs and associated labor-intensive procedures.

Considering the shortcomings of rodent models, there is an urgent need for high-throughput animal models that would speed up dramatically the early discovery phase of the search for novel and efficacious antifibrotics. Zebrafish offer a great opportunity in this regard, as they possess similar organs and tissues to those present in humans (Figure 1), and given the small size of the larvae and juveniles, they can be used for medium- to high-throughput platforms [9,10]. As a matter of fact, 70% of protein-coding human genes are related to genes found in zebrafish, and 82% of the genes known to be associated with human diseases have a zebrafish counterpart. To the best of our knowledge, an estimated percentage in fibrosis-related pathways or genes has not been reported, although some relevant pathways (such as TGFβ/Activin signalling pathway [11], MAPK/ERK signalling [12]) and genes (such as *tgf-β1*, [13] *2,* and *3* [11,13], *KRAS^G12D^* [14], *vinmentin* [11]) of fibrogenesis are active and expressed respectively in zebrafish. In addition, zebrafish have an immune system that is largely conserved with humans, with all different immune cell types present. Larvae possess a fully functional innate immune system with macrophages and neutrophils, whereas the adaptive immunity is only functionally mature from 4–6 weeks postfertilization on [15]. Moreover, unlike rodents, zebrafish can simply be immersed in compound-containing water, though other administration routes, including injection and oral gavage [16,17] can also be applied. Finally, zebrafish are genetically tractable and genome editing tools (i.e., CRISPR-Cas9 ribonucleoprotein complexes) as well as gene knockdown approaches using antisense oligonucleotides have frequently been used with success to obtain tailor-made models that mirror specific human diseases [18].

As a matter of fact, numerous disease-relevant compounds have been found using zebrafish-based assays [10], and in 2019 around ten compounds discovered in this way entered into early clinical trials [9]. For instance, ProHema, a derivative of prostaglandin E2 (PGE2), has been found to be capable of regulating the number of haematopoietic stem cells (HSCs) in adult zebrafish and has shown activity in Phase II clinical trials improving the engraftment of umbilical cord stem cells in leukaemia patients after transplantation [9,19].

However, one drawback of zebrafish is that, in contrast to most human organs (i.e., heart, pancreas, and kidney), their counterparts have been found to be regenerative and proliferative [20]. Despite this, zebrafish fibrotic models have been generated that closely mimic mammalian fibrotic models and clinical symptoms, especially with regard to the accumulation of extracellular matrix (ECM) [21,22,23]. Currently, there are only a handful of established zebrafish models of fibrosis, which highlights the opportunity for innovation and the possibility of developing and investigating mechanistically new models in detail.

Here, we give an overview of human fibrotic diseases, touch upon the underlying mechanisms known, and summarize the available zebrafish models of fibrotic disease, both larval and adult, generated within the past two decades. We focus on the modelling of organ-specific fibrotic diseases, such as cardiac fibrotic diseases, liver fibrotic diseases, pancreas fibrotic diseases, and others.

## 2. Fibrotic Diseases and Underlying Mechanisms

### 2.1. Fibrotic Diseases

From a mechanistic perspective, fibrosis is the result of an abnormal tissue repair that eludes homeostatic regulatory mechanisms, and then becomes a progressive fibrotic process that ultimately results in organ dysfunction and failure [24]. It is characterized by a massive net accumulation of extracellular matrix (ECM) that is composed of macromolecules, including collagens (e.g., COL1 and COL3), elastins, fibronectins, etc. Myofibroblast cells are considered to be responsible for the overproduction of ECM and fibrosis progression [25,26].

As a range of related disorders [25], fibrotic diseases have been classified into systemic fibrotic diseases (such as systemic sclerosis [27], and IgG4-associated tissue fibrosis [28]), organ-specific fibrotic diseases (such as cardiac, kidney, pulmonary, and liver fibrosis) [29], and other organ-specific fibrotic diseases (such as intestinal and bladder fibrosis). The most common organ-specific fibrotic diseases are listed in Table 1 [5,30,31,32].

### 2.2. Cellular Mechanism of Fibrosis

The progression of chronic diseases eventually resulting in fibrosis is the pathological outcome of a complex cross-talk between several key players like epithelial, endothelial and inflammatory cells that elicit and sustain fibrosis, and myofibroblasts that are the primary ECM-secreting cell type executing fibrosis.

It is generally accepted that repetitive injury to the epithelial compartment is a pivotal event in the development of fibrotic diseases. Indeed, epithelial cells can dedifferentiate upon continued stress into simplified and flattened cells that secrete paracrine factors like hedgehog and Wnt ligands, thereby stimulating myofibroblast differentiation. When cellular injury persists, these cells might even become senescent typically exhibiting apoptosis-resistance and displaying cell-cycle arrest without further proliferation and repair. Significantly, senescent cells also secrete numerous proinflammatory and profibrotic paracrine mediators (including TGF-b) that further drive the activation of myofibroblasts and amplify inflammatory processes [33,34,35].

During interstitial fibrosis, injury to the endothelial cells (together with the basal membrane delineating peritubular capillaries) can also result in capillary rarefaction, interstitial inflammation and fibroblast activation. The subsequent microvascular dysfunction can then further lead to local hypoxia, one of the forces causing fibrosis [36,37].

Inflammatory immune cells like macrophages are present in all types of diseases and fibrosis. They migrate to sites of cytokines produced (e.g., CCL2, CCL5) by injured resident cells. Significantly, macrophages can have a favourable role during acute injury as they promote wound healing. However, during prolonged inflammatory processes like CKD they might become profibrotic [38]. There are numerous indications that fibroblasts play an essential role in mediating fibrosis in organs. Activated fibroblasts [39] and myofibroblasts [40] originate from resident fibroblast populations and bone marrow (BM)-derived cells, and are able to produce various extracellular matrix (ECM) proteins such as procollagen and proteoglycans. As terminally differentiated cells, activated myofibroblasts are special fibroblasts that are rarely observed in non-pathological situation, but they are frequently observed in the process of wound healing. Activated myofibroblasts have been widely accepted to be the primary ECM-producing cells (especially COL1 and COL3 rich pathological ECM) [41]. These cells share some characteristics of smooth muscle cells and secrete α-smooth muscle actin (α-SMA), fibronectin ED-A, CD31, adhesion molecules and other mesenchymal cells markers [42].

In normal wound healing, most myofibroblasts undergo apoptosis and disappear following the completion of re-epithelialization [43,44,45]. However, persistent myofibroblast activation is a shared feature in fibrotic diseases. Therefore, overproduction of their hallmark, alpha-smooth muscle actin (α-SMA), is prevalently used as a marker of fibrosis [46]. 

Regarding the origin of fibrogenic myofibroblasts, most of them originate from resident cells, although it varies depending on the organ involved [47,48,49]. For instance, hepatic myofibroblasts in fibrotic liver are mostly derived from liver-resident hepatic stellate cells [48,50], mesothelial cells [51] and portal fibroblasts (PFs) [48]. A small contribution is made by bone marrow (BM)-derived cells (mesenchymal stem cells and fibrocytes) [48,50]. In the case of cardiac fibrosis, the source of myofibroblasts are resident fibroblasts and perivascular cells [37], while the contribution of other cell types such as endothelial cells, fibrocytes, epicardial cells, haematopoietic bone marrow-derived cells, is still controversial [47].

### 2.3. Molecular Mechanism of Fibrosis

A growing body of evidence suggests that numerous molecules, such as transforming growth factor-beta (TGF-b), Twist, Snai1, Wnt, Hedgehog and Notch are involved in regulating the various pathways of fibrogenesis. The most common outcome of their activity is an increasing population of activated myofibroblasts and the progressive development of fibrosis.

Among these molecules, TGF-b has been considered to be a primary mediator in the regulation of fibrosis, especially TGF-b1 that has been accepted as a master of myofibroblasts activation, transformation and differentiation in fibrosis [26,52]. It has been documented that TGF-b stimulates the EMT program in tubular epithelial cells, promoting the proliferation and activation of myofibroblasts, upregulated by increased *Twist* and *Snai1* gene levels [53,54]. 

As with organogenesis, the Wnt, Hedgehog (Hh) and Notch pathways also display essential roles in fibrogenesis [55,56]. Upregulation of numerous Wnt ligands has been detected in fibrotic tissue, leading to prolonged activation of the Wnt/β-catenin pathway. This protraction of β-catenin activation results in epithelial dedifferentiation and interstitial fibrosis [57,58]. Wnts have been established as playing an important role in myofibroblast activation and interstitial fibrosis which might be due to the essential role that Wnt ligands play in paracrine signalling between injured epithelial cells and interstitial myofibroblasts. Similarly, Hh ligands are also upregulated during fibrogenesis, accompanied by Hh pathway activation shown by increased *Gli1* gene expression [22]. This provides evidence that Hh ligands also stimulate myofibroblast activation through paracrine secretion in the signalling loop. 

In parallel to the Wnt and Hh pathways, the Notch pathway also plays a pivotal role in organs development and fibrogenesis. For instance, Notch induction was observed in cardiac fibrosis formation and regeneration along with the dedifferentiation of epithelial cells [59]. In addition, expression of Notch in cultured epithelial cells results in activation of their EMT program through regulating the gene expression of *Snail*, which is a key regulator of EMT [60]. Epithelial Notch may then be assumed to promote fibrosis in vivo, via activation of the EMT program. 

## 3. Zebrafish Models of Fibrotic Disease

### 3.1. Zebrafish Models of Cardiac Fibrosis

Cardiovascular disease (CD) is one of the leading causes of morbidity and mortality worldwide. Ischemic and many non-ischemic cardiomyopathies commonly involve major cardiac muscle loss and fibrosis, and are the primary causes of heart failure [61]. In adult mammals, dead cardiomyocytes, caused by myocardial infarction, are replaced with irreversible fibrotic scars rather than being regenerated [62]. To better understand the pathogenesis of CF, how regeneration affects fibrogenesis, and eventually seek a cure, some zebrafish models of CDs have been developed [21,63,64] in adult zebrafish, namely through ventricular apex resection and cryoinjury (Table 2).

#### 3.1.1. Cardiac Fibrosis Models by Ventricular Apex Resection (VAR)

VAR commences with anesthetizing and mounting of zebrafish, followed by incision of the chest wall and transection of the cardiac ventricle apex [72]. Massive ECM accumulation, detected with acid fuchsin orange G (AFOG), is observed in the wound of zebrafish 9 days post injury (dpi). Heart regeneration that is complete at 60 dpi starts with restoration of the ventricular wall in the wound area, followed by proliferation of myocytes [63,73,74].

A robust and fairly reproducible injury to the heart can be generated with this technique, and a certain part of the heart can be ablated specifically. The method is, however, inherently invasive, difficult to operate, low-throughput and sometimes causes lethality. Besides, not only are the cardiomyocytes resected, but also endocardial cells, epicardial cells, and vascular endothelial cells [72]. 

#### 3.1.2. Cardiac Fibrosis Models by Cryoinjury

Another surgery-dependent approach is cryoinjury. The procedure involves the rapid freezing and thawing of cardiac tissue for 20–25 s with a stainless steel cryoprobe precooled in liquid nitrogen. Cryoinjury is less lethal than ventricular resection, and fewer animals die after the surgery at 1 dpi [21]. Histologically, fish hearts that have undergone cryoinjury develop a collagen and fibrin-rich scar from 4 to 14 dpi, progressively dissolving the fibrin as well as contracting the scar area from 14 dpi on and taking around 2 months to heal [21,75]. Although functional recovery requires 6 months following the cryoinjury [76], the heart has a normal electrocardiogram at 30 dpi [11]. Additionally, compared to the VAR zebrafish model, collagen deposition in cryoinjury models was more pronounced, and a longer time was required for complete resorption [63,75]. Finally, using a multiple-cryoinjury approach it was demonstrated that the zebrafish heart shows a growing inefficacy in scar resorption related to the number of cryoinjuries applied. For instance, after six applications, the heart presented with uncomplete fibrotic tissue resolution and increased accumulation of collagen at the wound site. The phenotypic outcome was secondary to an enhanced recruitment of neutrophils and decreased proliferation and dedifferentiation of cardiomyocytes [65].

### 3.2. Zebrafish Models of Liver Fibrosis 

The liver is the largest internal organ in both mammals and lower vertebrates, such as zebrafish and frogs. Chronic liver damage is mainly caused by toxins, viral infections (e.g., hepatitis C virus (HCV)) [77], autoimmune conditions, and metabolic and genetic diseases [78]. Liver fibrosis is the common outcome of chronic or iterative insults.

Physiologically, with the exception of Kupffer cells, the zebrafish liver encompasses the same primary cell types (e.g., hepatocytes, stellate cells, biliary cells, and endothelial cells) [79] performing similar functions as their mammalian counterparts [80]. Therefore, zebrafish are considered to be an important tool for studying liver diseases. Several zebrafish fibrotic liver models have been documented using both larvae and adults, including alcoholic liver disease (ALD), non-alcoholic fatty liver disease (NAFLD) chemically induced models, and genetic models (Table 2). 

#### 3.2.1. Alcoholic Liver Disease (ALD)

Alcohol abuse is a common cause of liver fibrosis known as ALD. Ethanol (EtOH) immersion has been applied to both zebrafish larvae [68,81] and adults [66,82] for fibrotic liver modelling. To that end, zebrafish are immersed in EtOH-containing water replenished on a daily basis to keep the EtOH concentration stable and the housing environment clean. ECM accumulation was observed (visualized with Sirius Red staining) following 4 weeks (0.2% EtOH) [66] and 3 months (1% EtOH) [82] of treatment respectively, and the animals in both studies also developed steatosis and a hepatocyte ballooning phenotype. 

Importantly, ethanol treatment of genetically engineered zebrafish that express a hepatocyte-specific ablation system, can exacerbate and accelerate dramatically fibrogenesis compared to wild type zebrafish [67]. The ablation critically depends on the cell-specific expression of nitroreductase (NTR) that converts the nontoxic metronidazole (MTZ) in which the zebrafish are immersed, into a highly toxic DNA inter-strand cross-linking agent. For instance, by combining simultaneous NTR-mediated hepatocytes ablation and EtOH (1.5%, *v*/*v*) treatment, Huang et al. (2016) detected excessive overproduction and accumulation of the ECM protein COL1A with immunostaining after 25 h and 50 h in larvae. Similarly, COL1A deposition could also be detected in adult fish (1% EtOH) already at 48–72 h [67], much more quickly than the one to several months of exposure time that are typically needed with EtOH treatment alone [66,82]. However, although the combined treatment efficiently caused ECM overproduction, advanced fibrosis was not observed [67]. 

#### 3.2.2. Non-Alcoholic Fatty Liver Disease (NAFLD)

NAFLD is a much more common chronic liver disease than ALD. It is well known that NAFLD is associated with type 2 diabetes [83], obesity, metabolic syndrome [84], and some cardiovascular diseases [85]. Although many of these diseases have been modelled in mice, and a few zebrafish models of NAFLD were generated [69,86,87], only one zebrafish obesity-associated fibrotic liver model is described so far [69]. Investigating whether menopause is associated with the severity of liver fibrosis, Turola et al. [69] overfed male and female adults for 24 weeks with a high-calorie diet. Fish in all conditions, both male (young and old) and female (young and old), developed collagen deposition. Of interest, old female fish with failing ovarian function presented livers with the most severe fibrosis accumulation, and old male fish showed a higher degree of fibrosis in comparison to young male fish.

#### 3.2.3. Chemically Induced Liver Fibrosis Models

Carbontetrachloride (CCl4) and thioacetamide (TAA) are the most commonly used toxic chemicals to generate fibrotic liver disease through repetitive injection in rodents [88,89,90,91]. However, liver fibrosis was not detected in zebrafish larvae following CCl4 treatment using gene expression as a read-out [23]. In contrast, three days of TAA exposure induced ECM accumulation in the liver as visualized by Sirius-red staining and upregulated fibrosis-related genes (*col1a1*, *acta-2*, *hand-2*, *tgfb*) in zebrafish larvae [23]. Interestingly, the ECM accumulation in zebrafish larvae was visibly different from that found in human or rodent liver. This might be due to the difference in cell organization within the liver of fish and humans, e.g., the lack of lobular architecture, or less organized bile duct hepatocytes and stellate cells [79].

#### 3.2.4. Genetic Liver Fibrosis Models

Recently, a zebrafish model of fibrotic liver disease was reported, generated by overexpression of TGF-β1α (the counterpart of TGF-β1 in mammals) in the liver driven by the organ-specific *fabp10* promoter [13]. TGF-β1 has a critical role in the epithelial to mesenchymal transition (EMT) process and is involved not only in chronic lung [92] and kidney diseases [93], but also in chronic liver and cardiovascular diseases [11]. Moreover, it is generally upregulated in a range of fibrotic diseases [94]. Following 3–6 weeks of TGF-β1α overexpression using a 1 µM mifepristone-inducible system, abundant collagen accumulation was revealed in the liver in adult zebrafish by Sirius Red staining. The levels of accumulated collagen were reduced when fish were exposed to higher concentrations of mifepristone (2–3 µM) for 6 weeks. This reduction was demonstrated to correlate positively with an increased tumour progression [13].

Besides fibrogenic gene overexpression, mutations in mannose phosphate isomerase (*mpi*) present in hepatocytes also promotes hepatic fibrosis. According to a study by DeRossi et al. [68] *mpi* depletion in heterozygous adult zebrafish liver resulted in a continuous upregulation of fibrogenic genes (i.e., *col1a1a*, *col1a1b,* and *acta2*) and accumulation of collagen as detected with Masson’s trichrome staining. Of interest, mild *mpi* deletion in very early-stage zebrafish embryos (from 96 to 120 hpf) reinforced the fibrosis effects of EtOH administration and elevated the expression of *col1a1a* and *acta2* within 24 h. 

Based on the models developed to date, it can be concluded that advanced fibrosis in adult zebrafish requires prolonged and sustained injury [13,66,69,82]. Although fibrogenesis of the liver has been detected in larval zebrafish, sensitive techniques (e.g., IHC, qPCR, etc.) were required due to the lower abundancy of accumulated ECM [23,67,68]. Future model optimization should be directed towards a balance between the time length of treatment and ECM occurrence.

### 3.3. Zebrafish Models of Pancreas Fibrosis

Common diseases related to injury of the pancreas are diabetes mellitus, pancreatitis, and pancreatic adenocarcinoma. The zebrafish pancreas develops and functions to secrete hormones for energy homeostasis in the early embryo stage (≤48 hpf). Human pancreas- related diseases have been mimicked in zebrafish, namely chronic pancreatitis [22], cystic fibrosis [95], diabetes [96], and pancreatic cancer [14,97].

Fibrosis has been examined in some of these models. Jung and colleagues [22] overexpressed both Indian Hedgehog (*Hh*) [*Tg(Ptf1a-Gal4/UAS:GFP-UAS: Ihha)*] or Sonic *Hh* [*Tg(Ptf1a-Gal4/UAS:GFP-UAS: Shha)*] in zebrafish (Table 2). Both types of *Hh* overexpressing transgenic zebrafish exhibited identical phenotypes, i.e., Indian and Sonic *Hh* caused progressive pancreatic fibrosis in older animals. According to histopathologic analysis, *Hh*-induced progressive pancreatic fibrosis (manifested as the destruction of the histo-architecture) was observed in one-month-old fish. Moreover, ECM deposition was detected in 4-month-old fish using Masson’s trichrome staining. Meanwhile, an increased expression of α-SMA and TGF-β1 was detected [22,98].

Contrary to *Hh* transgenic fish, oncogenic *KRAS^G12D^* expression in the *elastase3I* domain resulted ultimately in pancreatic endocrine tumours [14]. This expression can be placed under the control of the zebrafish *ubb* promoter by using the construct *Tg(ubb-Lox-Nucleus-mCherry-Lox-GFP-KRAS^G12D^)*. The results show that the invasion of carcinoma stimulated fibrosis, as evidenced by the ECM accumulation visualized by trichrome staining.

### 3.4. Other Zebrafish Models of Fibrosis

Gill fibrosis and muscle fibrosis have also been described [70,71]. As the zebrafish counterparts of mammalian lungs, gills are sites for gas transfer and are important locations for chemoreception or gas sensing. Both lungs and gills are respiratory organs responsible for O_2_ uptake and CO_2_ excretion, and they share similar morphological features. Oh and colleagues [70] demonstrated a fibrotic gill response after exposure of adult fish to polyhexamethylene guanidine phosphate (PHMG-P), a pulmonary toxic compound used in humidifier disinfectants. Following persistent exposure over 28 days, gill fibrosis was evidenced both at the mRNA level (detected with qPCR) and protein level (detected with Masson’s trichrome staining) (Table 2).

In addition, since fibrosis is a major complication of ionizing irradiation exposure [99,100], Epperly et al. [100] modelled irradiation-induced fibrosis in zebrafish to address the current lack of models for screening of novel irradiation mitigators and protectors (Table 2). Using 30 Gy irradiation, 25% of fish developed abnormalities in the shape and structure of fin and tail, and massive ECM accumulation was detected in their dorsal musculature. Interestingly, following continuous treatment with a small molecule, ethyl pyruvate, the survival rate improved, and deposited collagen was reduced.

## 4. Discussion

Zebrafish (Danio rerio) are small vertebrates with highly conserved physiology to humans and with a high degree of conservation to the human genome also with respect to pharmaceutical drug targets. As a matter of fact, zebrafish larvae are amenable to high- throughput compound screening, both phenotypically [101,102] and even histologically [103]. By combining these features with easy handling and speed, they have emerged as a cost-efficient and valid alternative for disease modelling and large-scale drug screening over the last decade. Recent studies also show that the zebrafish is a suitable and cost-effective model for studying fibrosis. 

Despite the clear added value of the zebrafish model, there remain some challenges associated with its use as a fibrotic model. For instance, it is commonly accepted that regeneration accompanies fibrosis following injury in organs, such as the heart [21,63] and pancreas [96] in zebrafish. Since this high regeneration capacity, especially in case of larvae, is one of the major limitations to the development of fibrotic disease models, attenuating regeneration is of interest to favour fibrogenesis. A reduction in the regenerative capability is likely to be achieved through knocking down (KD) or knocking out (KO) the expression of regeneration-related genes. For instance, the KD or KO of some of these genes, such as telomerase [104] and Fos-related antigen 1 (Fra-1) [105], exacerbated bleomycin-induced pulmonary fibrosis in mice.

Furthermore, ECM accumulation in larvae is not always visible by histological staining (i.e., Sirius red, trichrome, and AFOG staining). This might be due to the small amount of ECM generated, which is much lower than in adult fish [13,23,67]. Variations in morphological architecture due to species differences can also be a barrier for fibrotic modelling in zebrafish [106]. Collagen deposition or fibrosis markers (such as TGF-β and α-SMA) must therefore be detected with more sensitive techniques, such as IHC or qPCR, thereby losing the throughput advantage over rodents. Instead, adult fish can be used, but these will not fully preserve the strengths of zebrafish larvae, i.e., they are less time-effective (zebrafish need 3 months to become adults), are not transparent (pigments interrupt the observation of fluorescence-highlighted organs), and are only low- to medium-throughput.

To summarize, recent studies show that zebrafish fibrotic models represent a promising and cost-effective alternative to rodent models. Zebrafish can recapitulate the pathophysiology of human organs due to a high level of genetic conservation, and similar organ morphologies and functions. It is believed that further technical developments and characterization of zebrafish models of fibrotic diseases will bring new insights into the molecular and cellular mechanisms underlying disease pathogenesis, thereby providing numerous opportunities for the identification and validation of new therapeutic targets and treatments.

## Figures and Tables

**Figure 1 ijms-22-06404-f001:**
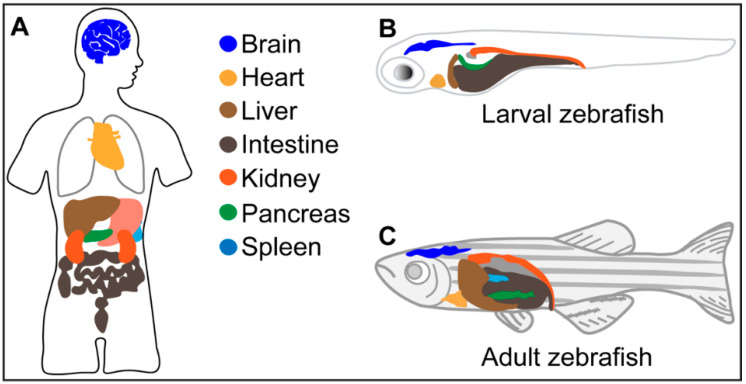
Corresponding organs and tissues in humans (**A**), and larval (**B**) and adult (**C**) zebrafish.

**Table 1 ijms-22-06404-t001:** Common human fibrotic diseases (adapted from Piera-Velazquez et al. [25]).

Organ-Specific Fibrotic Diseases
Cardiac Fibrosis
Pressure Overload Post-myocardial-infarction Chagas Disease-induced fibrosis
Kidney Fibrosis
Diabetic and Hypertensive Nephropathy Urinary Tract Obstruction-induced Kidney Fibrosis Inflammatory/Autoimmune-induced Kidney Fibrosis Aristolochic acid Nephropathy Polycystic Kidney Disease
Pulmonary Fibrosis
Idiopathic Pulmonary Fibrosis Silica-induced Pneumoconiosis (Silicosis) Asbestos-induced Pulmonary Fibrosis (Asbestosis) Chemotherapeutic Agent-induced Pulmonary Fibrosis
Liver and Portal Vein Fibrosis
Alcoholic and Non-Alcoholic Liver Fibrosis Hepatitis C-induced Liver Fibrosis Primary Biliary Cirrhosis Parasite-induced Liver Fibrosis (Schistosomiasis)
Intestinal Fibrosis Bladder Fibrosis Radiation-induced Fibrosis (various organs) Peritoneal Sclerosis Localized Scleroderma, Diffuse Fasciitis, and Keloids Dupuytren’s Disease Peyronie’s Disease Myelofibrosis Oral Submucous Fibrosis

**Table 2 ijms-22-06404-t002:** Overview of reported zebrafish models of fibrotic disease, differentiating the larval and adult stage, and techniques used for induction and validation.

	Larvae			Adults		
Organs	Induction	Detection	References	Induction	Detection	References
Heart	Not known	Not known	Not known	Ventricular apex resection (20%)	AFOG staining	Poss 2002 [63]
				Cryoinjury	Aniline Blue or MT staining	Chablais et al., 2011 [21]
				Multiple cryoinjuries	AFOG staining	Bise et al., 2020 [65]
Liver	TAA induction	Sirius Red staining & qPCR	van der Helm et al., 2018 [23]	EtOH induction	Picrosirius Red staining	Park and Kim 2019 [66]
	EtOH & NTR/MTZ ablation	IHC staining	Huang et al., 2016 [67]	EtOH & NTR/MTZ ablation	IHC staining	Huang et al., 2016 [67]
	EtOH & *mpi* knockdown	qPCR	DeRossi et al., 2019 [68]	EtOH & *mpi* knockdown	MT staining	DeRossi et al., 2019 [68]
				Ovarian senescence & obesity	Sirius Red staining	Turola et al., 2015 [69]
				Overexpressed tgfbβ1α induction	Sirius Red & IHC staining	Yan et al., 2019 [13]
Pancreas	Not known	Not known	Not known	Hedgehog (Hh)-induction	MT staining	Jung et al., 2011 [22]
				Transgene *KRAS^G12D^* expression	MT staining	Oh and Park 2019 [14]
Other organs (tissues)	Not known	Not known	Not known	PHMG-P induced gill fibrosis	MT staining	Oh et al., 2018 [70]
				Ionizing irradiation caused muscle fibrosis	MT staining	Epperly et al., 2012 [71]

NTR, nitroreductase; MTZ, metronidazole; AFOG, acid fuchsin orange G; MT staining, Masson’s trichrome staining; TAA, thioacetamide; IHC, immunochemistry; *mpi*, mannose phosphate isomerase; PHMG-P, polyhexamethylene guanidine phosphate.

## Data Availability

Data sharing not applicable. No new data were created or analyzed in this study. Data sharing is not applicable to this article.

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
