# Peer review of "Using Zebrafish as a Disease Model to Study Fibrotic Disease"

_ijms, 2021, doi:10.3390/ijms22126404_

Round 1
Reviewer 1 Report
To understand the pathogenesis and investigate novel therapeutic interventions several in vitro and in vivo model systems for fibrogenesis have been used. These include in vivo mouse and rat models, that have been proven valuable models, but are expensive and time consuming.
Zebrafish embryos are often used to perform high throughput drug screens and in this review Wand and coworkers try to summarize the zebrafish models used to study different fibrotic disorders including various cardiovascular diseases (CVD), liver cirrhosis and chronic kidney disease (CKD).
I recommended this paper for publication and I only have some suggestions and some minor concerns listed below.
Finally, I suggest that authors comment and discuss the data presented in other papers, in order to compare the results of this analysis with those known in other contexts.
Here are just a few examples of useful references.
Authors must to cite and discuss in the section ” 3.1.2. Cardiac fibrosis models by cryoinjury” page 7, the recent paper of Muse et al., 2021: Multiple cryoinjuries modulate the efficiency of zebrafish heart regeneration. Scientific Reports | (2020) 10:11551.
Add also such information in the Table 2.
Authors must cited as diet-induced obesity (DIO) modeling of NAFLD in zebrafish the paper of Forn-Cuní et al., 2015. Liver immune responses to inflammatory stimuli in a diet-induced obesity model of zebrafish. J Endocrinol. 2015 Feb;224(2):159-70.
Add this information in the section 3.2.2 and Table 2 and change the sentence “only one zebrafish obesity-associated fibrotic liver model is described so far” because the paper of For-Cunì describe another fibrotic liver model in zebrafish.
In the section “3.3. Zebrafish Models of Pancreas Fibrosis” line 307 together with reference number 90, authors must add citation of Schiavone et al. 2014 entitled” Schiavone M, Rampazzo E, Casari A, Battilana G, Persano L, Moro E, Liu S, Leach SD, Tiso N, Argenton F. Zebrafish reporter lines reveal in vivo signaling pathway activities involved in pancreatic cancer. Dis Model Mech. 2014 Jul;7(7):883-94”.
Reviewer 2 Report
This is a very interesting topic since animal models will always be an important selection of available experimental model in medicine. Since fibrosis is a complex process involving many different cells, not easily modelled in cell culture. This is a nice overview of different fibrotic models. However, there are some aspects of zebrafish models that were not discussed or commented on in the manuscript but could increase the impact of the review.
The authors stated that and 82% of the genes known to be associated with human diseases have a zebrafish counterpart. Is there an estimate of the percentage in fibrosis-related pathways or genes? And are there any missing key proteins?
The immune system also plays an important role in fibrosis, how similar is it in zebrafish to humans in respect to fibrosis?
How relevant is the zebrafish model in view of drug testing?
Page 5, line 128-129, you state that fibroblasts can originate from leukocytes, but this is not described in the attached references. By which study was this supported?
Page 5, line 145-146, the HSC abbreviation probably stands for hepatic stellate cells, which are the source of the myofibroblast in the liver.
